# Dynamic Batch Norm Statistics Update for Natural Robustness

## Abstract

DNNs trained on natural clean samples have been shown to perform poorly on corrupted samples, such as noisy or blurry images. Various data augmentation methods have been recently proposed to improve DNN's robustness against common corruptions. Despite their success, they require computationally expensive training and cannot be applied to off-the-shelf trained models. Recently, updating only BatchNorm (BN) statistics of a model on a single corruption has been shown to improve its accuracy on that corruption significantly. However, adopting the idea at inference time when the type of corruption changes decreases the effectiveness of this method. In this paper, we harness the Fourier domain to detect the corruption type, a challenging task in the image domain. We propose a unified framework consisting of a corruption-detection model and BN statistics update that can improve the corruption accuracy of any off-the-shelf trained model. We benchmark our framework on different models and datasets. Our results demonstrate about 8% and 4% accuracy improvement on CIFAR10-C and ImageNet-C, respectively. Furthermore, our framework can further improve the accuracy of state-of-the-art robust models, such as AugMix and DeepAug.

## 1 Introduction

Deep neural networks (DNNs) have been successfully applied to solve various vision tasks in recent years. At inference time, DNNs generally perform well on data points sampled from the same distribution as the training data. However, they often perform poorly on data points of different distribution, including corrupted data, such as noisy or blurred images. These corruptions often appear naturally at inference time in many real-world applications, such as cameras in autonomous cars, x-ray images, etc. Not only DNNs' accuracy drops across shifts in the data distribution, but also the well-known overconfidence problem of DNNs impedes the detection of domain shift.

One straightforward approach to improve the robustness against various corruptions is to augment the training data to cover various corruptions. Recently, many more advanced data augmentation schemes have also been proposed and shown to improve the model robustness on corrupted data, such as SIN Geirhos et al. (2018a), ANT Rusak et al. (2020a), AugMix Hendrycks et al. (2019), and DeepAug Hendrycks et al. (2021). Despite their effectiveness, these approaches require computationally expensive training or re-training process.

Two recent works (Benz et al., 2021; Schneider et al., 2020) proposed a simple batch normalization (BN) statistics update to improve the robustness of a pre-trained model against various corruptions with minimal computational overhead. The idea is to only update the BN statistics of a pre-trained model on a target corruption. If the corruption type is unknown beforehand, the model can keep BNs updating at inference time to adapt to the ongoing corruption. Despite its effectiveness, this approach is only suitable when a constant flow of inputs with the same type of corruption is fed to the model so that it can adjust the BN stats accordingly.

In this work, we first investigate how complex the corruption type detection task itself would be. Although corruption type detection is challenging in the image domain, employing the Fourier domain can make it much more manageable because each corruption has a relatively unique frequency profile. We show that a very simple DNN can modestly detect corruption types when fed with a specifically normalized frequency spectrum.

Given the ability to detect corruption types in the Fourier domain, we adopt the BN statistic update method such that it can change the BN values dynamically based on the detected corruption type. The overall architecture of our approach is depicted in Figure 1. First, we calculate the Fourier transform of the input image, and after applying a specifically designed normalization, it is fed to the corruption type detection DNN. Based on the detected corruption, we fetch the corresponding BN statistics from the BN stat lookup table, and the pre-trained network BNs are updated accordingly. Finally, the dynamically updated pre-trained network processes the original input image.

In summary, our contributions are as follows:

- We harness the frequency spectrum of an image to identify the corruption type. On ImageNet-C, a shallow 3-layer fully connected neural network can identify 16 different corruption types with $65.88\%$ accuracy. The majority of the misclassifications occur between similar corruptions, such as different types of noise, for which the BN stat updates are similar nevertheless.

- Our framework can be used on any off-the-shelf pre-trained model, even robustly trained models, such as AugMix Hendrycks et al. (2019) and DeepAug Hendrycks et al. (2021), and further improves the robustness.

- We demonstrate that updating BN statistics at inference time as suggested in (Benz et al., 2021; Schneider et al., 2020) does not achieve good performance when the corruption type does not continue to be the same for a long time. On the other hand, our framework is insensitive to the rate of corruption changes and outperforms these methods when dealing with dynamic corruption changes.

## 2 METHOD

### 2.1 OVERALL FRAMEWORK

The overview of our framework is depicted in Figure 1. It consists of three main modules: A) a pre-trained model on the original task, such as object detection, B) a DNN trained to detect corruption type, and C) a lookup table storing BN statistics corresponding to each type of corruption. This paper mainly focuses on improving the natural robustness of trained DNNs. However, the framework can be easily extended to domain generalization and circumstances where the lookup table may update the entire model weights or even the model architecture itself.

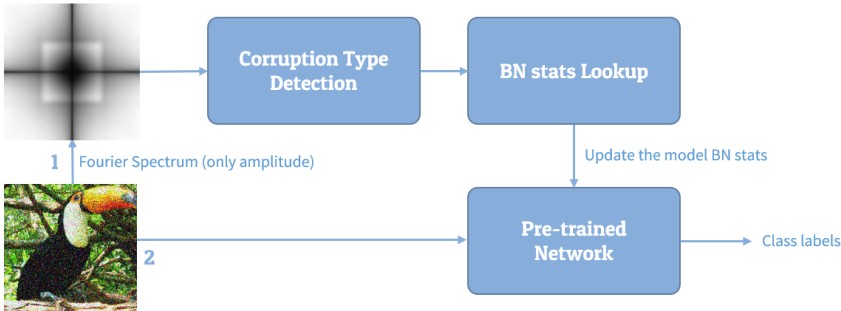

Figure 1: Overall Framework

### 2.2 ADAPTATION TO NEW CORRUPTIONS

In (Benz et al., 2021; Schneider et al., 2020), a simple BN statistic update has significantly improved the natural robustness of trained DNNs. Figure 2 show the effectiveness of their approach on various corruption types. The drawback of their approach is that the BN statistics obtained for one type of corruption often significantly degrades the accuracy for other types of corruption, except for similar corruption, such as different types of noise. The authors claim that in many applications, such as autonomous vehicles, the corruption type will remain the same for a considerable amount of time. Consequently, the BN statistics can be updated at inference time. However, neither of those papers

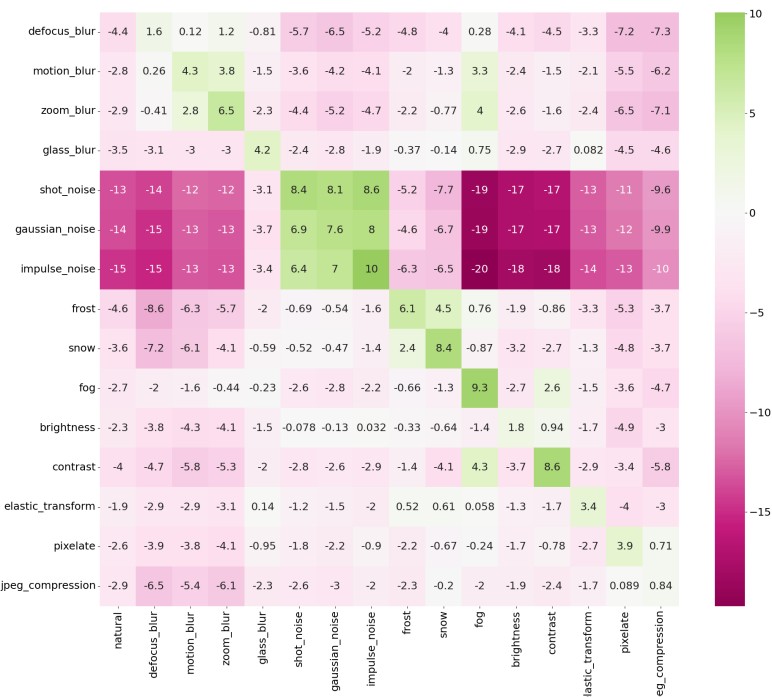

Figure 2: ResNet18 (ImageNet-C): The y-axis shows the corruption with which the model BN stats are updated. The x-axis shows the corruption on which the model performance is evaluated. The numbers in the cells are accuracy gain compared to the original model, the model with BN stats obtained from the natural dataset.

has shown the performance of BN statistic update when the corruption type changes. We conduct an experiment in Section 3.4 to show that detecting corruption types and utilizing appropriate BN stats provides better results when the corruption type is not fixed.

## 2.3 CORRUPTION DETECTION

The average Fourier spectrum of different corruptions has been shown to have different visual appearances Yin et al. (2019). However, conducting a corruption classification on the Fourier spectrum of individual images is not a trivial task. Feeding a DNN with the raw Fourier spectrum leads to poor results and unstable training. Here, we first visually investigate the Fourier spectrum of various corruption types. Then, we propose a tailored normalization technique and a shallow DNN to detect corruption types.

We denote an image of size $(d_1, d_2)$ by $x \in R^{d_1 \times d_2}$. We omit the channel dimension here because the Fourier spectrum of all channels turns out to be similar, when the average is taken over all samples. We only show the results of the first channel here. We denote natural and corrupted data distribution by $D_n$ and $D_c$, respectively. We denote 2D discrete Fourier transform operation by $F$. In this paper, we only consider the amplitude component of $F$ since the phase component does not help much with corruption detection. Moreover, we shift the low-frequency component to the center for better visualization.

Figure 3 shows the normalized Fourier spectrum of different corruption types in CIFAR10-C. The results on ImageNet-C is presented in Figure 4. We explain the normalization process in the next paragraph. For visualization purposes, we clamp the values above one. However, we do not clamp pixel values of the input when fed to the corruption detection model. As shown in Figure 3, most corruption types have a distinguishable average Fourier spectrum. The almost identical ones, i.e., different types of noise, are not needed to be distinguished accurately because the BN stat updates for one of them can improve the accuracy for others nevertheless, as shown in Figure 2.

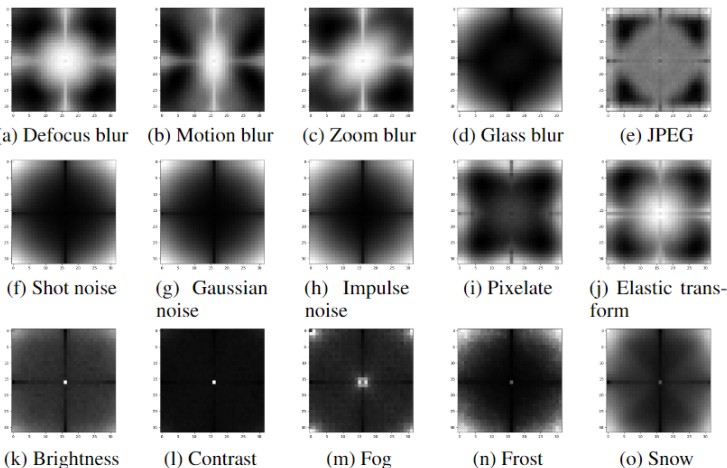

Figure 3: Normalized Fourier spectrum of CIFAR10-C dataset

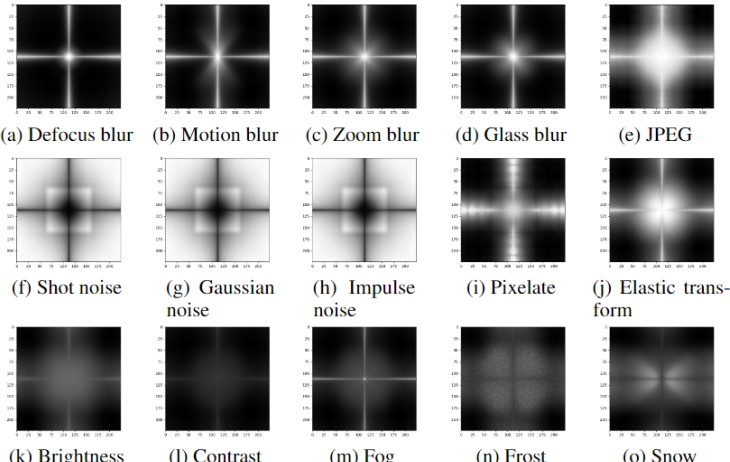

Figure 4: Normalized Fourier spectrum of ImageNet-C dataset

To normalize the data, we first obtain the average Fourier spectrum of the natural samples, denoted by $\epsilon_n = \mathbb{E}_{x \sim D_n}[||F(x)||]$. Then we compute normalized Fourier spectrum by $log(\frac{\mathbb{E}_{x \sim D_c}[||F(x)||]}{\epsilon_n} + 1)$ for each corruption type, separately. For corruption detection purpose, we substitute the expected value over the entire corruption type dataset by an individual image, i.e., $log(\frac{|F(x)|}{\epsilon_n} + 1)$. We empirically find this specific normalization to outperform others significantly. The intuition behind this normalization is twofold: First, natural images have a higher concentration in low frequencies Yin et al. (2019). Although corrupted images also have large values on low-frequency components, they may also have large concentration on high-frequency components, depending on the corruption. Hence, we divide the values by $\epsilon_n$ to ensure that model does not exclusively focus on low-frequency components during training. Second, the range of values from one pixel to another may vary multiple orders of magnitude, which causes instability during training. Typical normalization techniques on unbounded data, such as tanh or sigmoid transforms, leads to poor accuracy because values larger than a certain point converge to 1 and become indistinguishable.

We employ a three-layer fully connected (FC) neural network for corruption-type detection. Despite having an image-like structure, we avoid using convolutional neural networks (CNNs) here because of the apparent absence of shift-invariance in the Fourier spectrum. Due to the symmetry in the Fourier spectrum, we only feed half of the Fourier spectrum to the model. For CIFAR10, we flatten the 2D data and feed it to a three-layers FC model with 1024, 512, and 16 neurons. Note that this

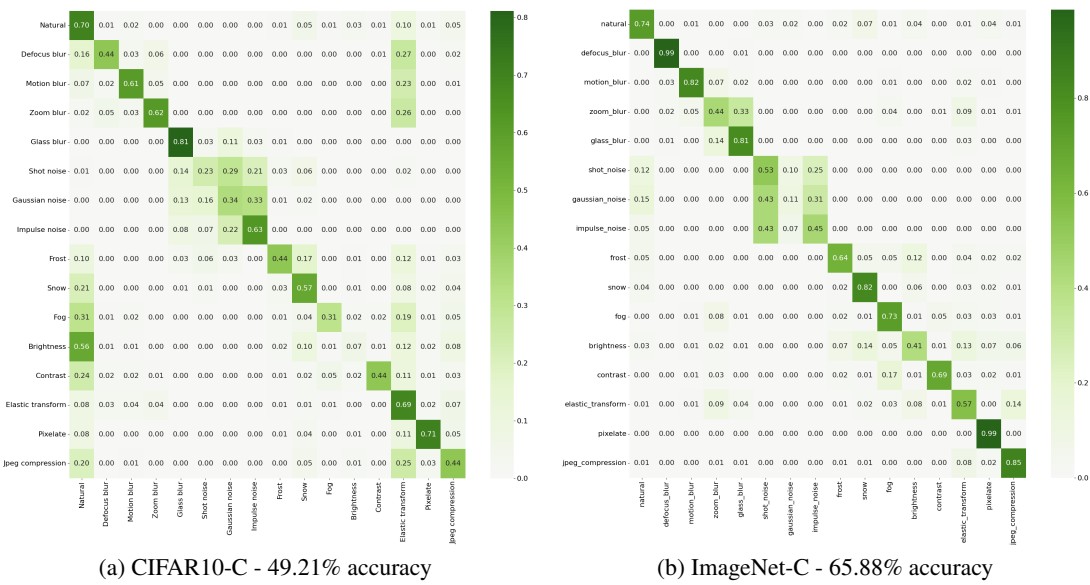

(a) CIFAR10-C - 49.21% accuracy        (b) ImageNet-C - 65.88% accuracy

Figure 5: Corruption type detection model's confusion matrix

paper deals with 15 corruption types and natural data as previously experimented in (Benz et al., 2021; Schneider et al., 2020). For ImageNet-C, we first use 2D average pooling with kernel size and stride of 2 to reduce the input size. Then, we flatten the output and feed it to a model with three FC layers of size 2058, 512, and 16. Additionally, we use ReLU function as non-linearity after the first and second layers.

We train the model with stochastic gradient descent (SGD) for 50 epochs. We decrease the learning rate by a factor of 10 at epochs 20 and 35. We only use a small number of samples for training, i.e., 100 samples per corruption/intensity, and we keep the rest for validation. Using the Fourier spectrum and the proposed normalization method, we achieve validation accuracy of 49.21% and 65.88% on CIFAR10-C and ImageNet-C, respectively. The same architecture and model capacity only yields 7.64% and 6.32% accuracy in the image domain. We also could not achieve good accuracy with CNNs in the image domain. The confusion matrix of the corruption detection is presented in Figure 5.

## 3 EXPERIMENTAL SETUP

**Datasets & Metrics.** CIFAR10 dataset Krizhevsky et al. (2009) contains $32 \times 32$ color images of 10 classes, with 50,000 training samples and 10,000 test samples. ImageNet dataset Deng et al. (2009) contains around 1.2 millions images of 1000 classes. For ImageNet, we resize images to $256 \times 256$ and take the center $224 \times 224$ as input. CIFAR10-C and ImageNet-C datasets Hendrycks & Dietterich (2019) contain corrupted test samples of the original CIFAR10 and ImageNet. There are 15 test corruptions and 4 hold-out corruptions. For a fair comparison with previous work, we only use the 15 test corruptions as in (Benz et al., 2021; Schneider et al., 2020). Each corruption type, $c$, contains 5 different intensities or severity level, denoted by $s$. Similar to Hendrycks et al. (2019), we use unnormalized corruption error $uCE = \sum_{s=1}^{5} E_{c,s}$ on CIFAR10, and normalized corruption error $CE = \sum_{s=1}^{5} E_{c,s} / \sum_{s=1}^{5} E_{c,s}^{AlexNet}$ for ImageNet-C. Corruption error averaged over all 15 corruptions is denoted by $mCE$.

**Models.** Our framework consists of two DNNs, namely the corruption type detector and a pre-trained model on the original task. The details of the corruption type detector model are explained in Section 2.3. For CIFAR10, we consider ResNet-20, ResNet-110 He et al. (2016), VGG-19 Simonyan & Zisserman (2014), WideResNet-28-10 Zagoruyko & Komodakis (2016), and DenseNet (L=100, k=12)

Table 1: Evaluation results on CIFAR10-C

| Model | All combined (accuracy) | | | On natural images (accuracy) | | | On corrupted images (accuracy) | | | On corrupted images (mCE) | | |
|---|---|---|---|---|---|---|---|---|---|---|---|---|
| - | Base | Ours | Δ | Base | Ours | Δ | Base | Ours | Δ | Base | Ours | Δ |
| ResNet-20 | 69.46% | 76.82% | 7.37% | 91.62% | 90.86% | -0.76% | 67.98% | 75.89% | 7.91% | 32.02% | 24.11% | -7.91% |
| ResNet-110 | 72.06% | 80.59% | 8.53% | 93.55% | 93.01% | -0.54% | 70.63% | 79.76% | 9.13% | 29.37% | 20.24% | -9.13% |
| VGG-19 | 74.08% | 80.69% | 6.61% | 93.19% | 92.95% | -0.24% | 72.81% | 79.87% | 7.07% | 27.19% | 20.13% | -7.07% |
| WRN-28-10 | 78.00% | 85.21% | 7.22% | 96.23% | 96.05% | -0.18% | 76.78% | 84.49% | 7.71% | 23.22% | 15.51% | -7.71% |
| DenseNet | 74.34% | 81.91% | 7.58% | 95.04% | 94.49% | -0.55% | 72.96% | 81.07% | 8.12% | 27.04% | 18.93% | -8.12% |

Table 2: Per corruption accuracy on CIFAR10-C

| Model | Blur | | | | Noise | | | Weather | | | Digital | | | | |
|---|---|---|---|---|---|---|---|---|---|---|---|---|---|---|---|
| - | Defocus | Motion | Zoom | Glass | Shot | Gauss | Impulse | Frost | Snow | Fog | Brightness | Contrast | Elastic | Pixelate | JPEG |
| ResNet-20 | 75.95 | 68.46 | 68.96 | 45.67 | 55.00 | 43.11 | 55.73 | 70.65 | 75.28 | 82.73 | 89.67 | 67.98 | 77.66 | 67.88 | 74.95 |
| ResNet-20 (ours) | 83.80 | 78.15 | 80.31 | 58.66 | 68.28 | 62.35 | 66.50 | 79.28 | 77.35 | 85.73 | 89.43 | 77.93 | 79.79 | 76.12 | 74.65 |
| ResNet-110 | 79.83 | 73.60 | 73.67 | 43.30 | 55.10 | 42.19 | 56.33 | 73.18 | 78.26 | 87.10 | 91.82 | 75.71 | 81.18 | 71.12 | 77.02 |
| ResNet-110 (ours) | 86.64 | 82.30 | 83.73 | 62.44 | 73.96 | 68.41 | 70.31 | 82.78 | 80.96 | 89.10 | 91.67 | 84.47 | 82.72 | 79.54 | 77.34 |
| VGG-19 | 80.41 | 76.19 | 77.43 | 52.58 | 59.75 | 46.59 | 50.70 | 77.30 | 80.34 | 85.63 | 91.65 | 69.41 | 83.23 | 78.97 | 81.91 |
| VGG-19 (ours) | 86.77 | 83.47 | 84.55 | 64.57 | 74.12 | 67.29 | 63.93 | 84.26 | 82.39 | 87.95 | 91.75 | 77.67 | 84.72 | 83.05 | 81.58 |
| WRN-28-10 | 84.20 | 81.40 | 80.38 | 59.87 | 62.92 | 50.30 | 54.18 | 82.06 | 85.69 | 90.11 | 94.87 | 80.84 | 86.62 | 77.91 | 80.35 |
| WRN-28-10 (ours) | 90.43 | 87.62 | 88.44 | 71.77 | 78.43 | 72.89 | 71.63 | 88.78 | 87.67 | 92.37 | 94.79 | 88.53 | 87.72 | 85.17 | 81.08 |
| DenseNet | 82.16 | 78.54 | 75.52 | 54.83 | 56.73 | 44.44 | 46.42 | 78.79 | 82.73 | 88.77 | 93.20 | 79.77 | 84.23 | 71.29 | 76.90 |
| DenseNet (ours) | 87.93 | 85.09 | 84.82 | 68.32 | 72.56 | 65.80 | 65.87 | 85.78 | 85.53 | 90.53 | 93.16 | 86.98 | 84.85 | 82.00 | 76.85 |

Huang et al. (2017). All CIFAR10 models are adopted from a public *github* repository[1]. For ImageNet, we consider ResNet-18, ResNet-50 He et al. (2016), VGG-19 Simonyan & Zisserman (2014), WideResNet-50 Zagoruyko & Komodakis (2016), DenseNet-161 Huang et al. (2017). All ImageNet models are adopted from *torchvision* library Marcel & Rodriguez (2010). We also adopted trained ResNet-50 models from state-of-the-art robustness literature, i.e., Stylized ImageNet training (SIN) Geirhos et al. (2018a), adversarial noise training (ANT) Rusak et al. (2020a), AugMix Hendrycks et al. (2019), and DeepAug Hendrycks et al. (2021).

**BN Statistics.** In this paper, we specifically adopted BN stat update from Schneider et al. (2020) with parameters $N = 1$ and $n = 1$. For a corruption $c$, this choice of parameters indicates that we take an average of a natural BN stats and the BN stats of the corruption $c$. We compute BN stats from the same samples we use to train the corruption-type detection model. Due to the small sample size for BN stat adoption, we find that taking an average with natural BN stats leads to better results than only using the target corruption BN stats.

## 3.1 EVALUATION ON CIFAR10-C

Table 1 presents the results of CIFAR10-C over several models. Our approach improves the accuracy over all corruptions by around $8\%$. However, the accuracy over natural samples is dropped by less than $1\%$. Because the base model is trained on natural samples, any misclassification of natural samples in the corruption detection model negatively affects the model performance, while any correct classification of corruptions positively affects the accuracy. As shown in Table 2, our approach significantly improves the accuracy over all the corruption types, except for brightness and JPEG corruption, in which the accuracy barely changes. Note that these two corruptions have the least improvement when BN stat is applied, as shown in Figure 2.

## 3.2 EVALUATION ON IMAGENET-C

Evaluation results on ImageNet-C is shown in Table 3 and 4. We observe a similar pattern as CIFAR10 with a slightly smaller improvement. Here, accuracy improvement is around $4\%$. Similarly, improvement occurs over all corruptions except for brightness and JPEG.

---

[1]https://github.com/bearpaw/pytorch-classification

Table 3: Evaluation results on ImageNet-C

| Model | All combined (accuracy) | | | On natural images (accuracy) | | | On corrupted images (accuracy) | | | On corrupted images (mCE) | | |
|---|---|---|---|---|---|---|---|---|---|---|---|---|
| - | Base | Ours | Δ | Base | Ours | Δ | Base | Ours | Δ | Base | Ours | Δ |
| ResNet-18 | 34.06% | 37.94% | 3.88% | 69.76% | 68.57% | -1.18% | 31.68% | 35.90% | 4.22% | 86.63% | 81.48% | -5.15% |
| ResNet-50 | 40.48% | 44.42% | 3.94% | 76.13% | 75.04% | -1.09% | 38.10% | 42.38% | 4.28% | 78.43% | 73.13% | -5.29% |
| VGG-19 | 36.48% | 40.17% | 3.69% | 74.22% | 73.42% | -0.80% | 33.96% | 37.95% | 3.99% | 83.77% | 78.84% | -4.93% |
| WRN-50 | 44.33% | 47.20% | 2.87% | 78.47% | 77.03% | -1.44% | 42.06% | 45.21% | 3.15% | 73.31% | 69.50% | -3.81% |
| DenseNet | 47.57% | 50.08% | 2.52% | 77.14% | 76.56% | -0.58% | 45.59% | 48.32% | 2.72% | 68.88% | 65.53% | -3.35% |

Table 4: Per corruption accuracy on ImageNet-C

| Model | Blur | | | | Noise | | | Weather | | | | Digital | | | |
|---|---|---|---|---|---|---|---|---|---|---|---|---|---|---|---|
| - | Defocus | Motion | Zoom | Glass | Shot | Gauss | Impulse | Frost | Snow | Fog | Brightness | Contrast | Elastic | Pixelate | JPEG |
| ResNet-18 | 28.03 | 29.62 | 29.57 | 22.97 | 20.81 | 22.67 | 17.57 | 28.22 | 24.10 | 33.88 | 58.85 | 30.80 | 39.78 | 42.13 | 46.25 |
| ResNet-18 (ours) | 29.58 | 33.35 | 31.42 | 26.20 | 27.83 | 29.68 | 26.31 | 32.24 | 31.36 | 41.60 | 58.40 | 37.45 | 40.48 | 46.01 | 46.61 |
| ResNet-50 | 35.52 | 36.35 | 36.32 | 25.48 | 28.60 | 31.12 | 26.59 | 35.17 | 30.54 | 43.41 | 65.19 | 35.95 | 43.34 | 45.55 | 52.44 |
| ResNet-50 (ours) | 36.47 | 40.39 | 38.39 | 29.77 | 35.69 | 38.04 | 33.71 | 38.83 | 37.44 | 48.70 | 65.16 | 42.87 | 44.87 | 51.77 | 53.60 |
| VGG-19 | 28.63 | 31.44 | 31.08 | 20.94 | 25.19 | 27.56 | 23.71 | 32.10 | 28.35 | 40.02 | 62.45 | 32.73 | 37.75 | 39.36 | 48.07 |
| VGG-19 (ours) | 29.94 | 34.91 | 32.43 | 24.10 | 30.54 | 32.73 | 31.98 | 36.43 | 35.18 | 46.85 | 62.76 | 39.01 | 39.36 | 45.13 | 47.96 |
| WRN-50 | 39.35 | 40.08 | 38.48 | 28.55 | 35.88 | 37.98 | 33.12 | 38.65 | 32.35 | 44.53 | 67.47 | 38.45 | 46.16 | 52.92 | 56.87 |
| WRN-50 (ours) | 36.60 | 42.38 | 38.45 | 30.79 | 43.45 | 45.56 | 40.98 | 42.75 | 40.26 | 48.33 | 66.46 | 44.58 | 47.06 | 55.45 | 55.06 |
| DenseNet | 40.09 | 40.12 | 38.42 | 29.58 | 40.95 | 42.31 | 37.18 | 43.84 | 39.64 | 52.13 | 69.80 | 50.01 | 47.42 | 54.53 | 57.89 |
| DenseNet (ours) | 38.98 | 42.64 | 40.26 | 31.05 | 45.31 | 47.37 | 43.11 | 46.80 | 46.04 | 56.34 | 69.35 | 52.96 | 47.37 | 58.33 | 58.85 |

## 3.3 Evaluation on robust models

In this section, we investigate if our approach can further improve the accuracy of state-of-the-art models on ImageNet-C. Table 5 presents the evaluation of five state-of-the-art models. Our approach consistently improves the performance of robust approaches even further. Note that here we exclude the data we use to train the corruption type detection model from the validation set. That explains the small discrepancy between the base accuracy reported in the paper and those in previous work.

## 3.4 Inference Time Adaptation

Two recent papers (Benz et al., 2021; Schneider et al., 2020) that investigated BN statistics update suggested that the idea can be used at inference time, and the model will adopt to a new corruption eventually. However, they have never empirically evaluated their performance for inference time adaptation. Here, we start with the original model trained on clean samples. Then, during evaluation, after a certain number of batches, we randomly pick another corruption and then continue evaluating the model. The samples within one batch come from only a single corruption, and there are 16 samples in each batch. We let the model BN stats be updated from the last ten batches at the beginning of each batch. Because our approach does not update the BN stat lookup table, it is insensitive to how the inference time evaluation is conducted, and consequently, the performance is similar.

The results of the experiment are shown in Figure 6. In CIFAR10, only in VGG-19 and only when we let the corruption stay the same for 32 consecutive batches our approach is underperformed. In ImageNet, both VGG-19 and ResNet18 outperforms our approach only after 32 successive

Table 5: Accuracy of ResNet50 on ImageNet-C

| Model | Base | Our adaptation | Δ |
|---|---|---|---|
| resnet50 | 38.10% | 42.38% | 4.28% |
| resnet50 SIN Geirhos et al. (2018a) | 37.78% | 40.29% | 2.51% |
| resnet50 ANT+SIN Rusak et al. (2020a) | 46.63% | 48.71% | 2.08% |
| resnet50 AugMix Hendrycks et al. (2019) | 46.96% | 50.44% | 3.47% |
| resnet50 DeepAug Hendrycks et al. (2021) | 51.47% | 53.18% | 1.71% |
| resnet50 DeepAug+AugMix Hendrycks et al. (2021) | 55.67% | 59.12% | 3.45% |

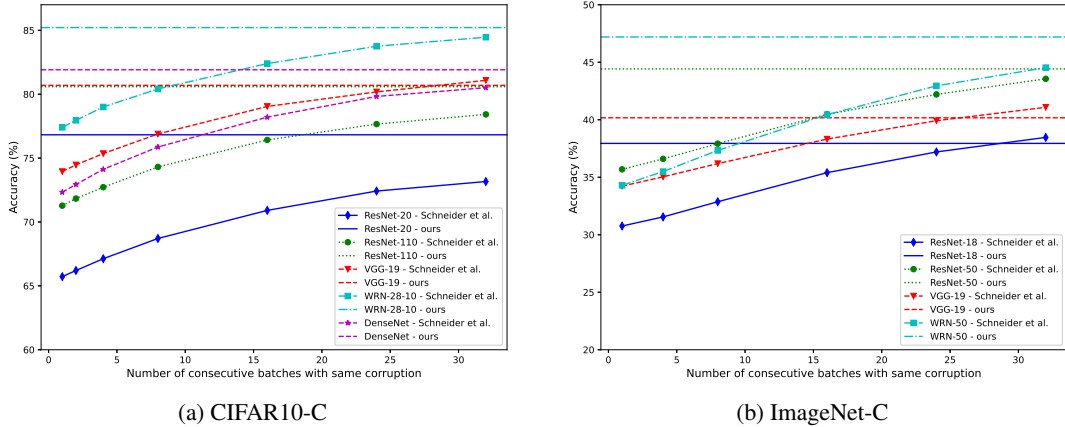

(a) CIFAR10-C              (b) ImageNet-C

Figure 6: BN stat update vs. our framework at inference time. The x-axis shows the number of consecutive batches for which the corruption remains the same during the evaluation. It takes many consecutive batches of the same corruption for models to catch up to the corruption change when inference time BN stat update is deployed. Our framework, however, is insensitive to the corruption changes and can adapt instantly.

batches. This experiment reveals that the original BN stat update mechanism in (Benz et al., 2021; Schneider et al., 2020) only works when input corruption remains the same for a considerable number of consecutive samples. Although this assumption is reasonable for some applications, such as autonomous vehicles with continuous stream input, it does not hold for many applications, particularly for non-stream inputs common in healthcare applications.

## 4    LIMITATIONS & DISCUSSION

One major limitation of the current framework is that it needs data samples from all corruption types to train the corruption type detection model. Although using the Fourier spectrum allows us to train the corruption detector easily with a small number of samples, it still limits the generalizability of the framework to unseen corruptions. One solution to this problem is to attach an off-the-shelf outlier detection mechanism or an uncertainty mechanism to discover new types of corruption at inference time. Then, we can make a new entry in the BN stat lookup table, and the model can gradually learn BN statistics at inference time by observing multiple samples from the new class. Hence, we can prevent the need to collect image samples from all corruptions during training. Another related perspective is to frame the *supervised* corruption type detection as an *unsupervised* problem. This reformulation is possible because the corruption labels themselves are nonessential in our framework. For example, we can use a clustering algorithm to cluster different corruption and then associate each cluster with an entry in the BN stats table. This strategy can also be extended to detect new clusters at inference time for better generalization. We will investigate this idea in future work.

In this paper, Our framework is only evaluated on natural and corrupted images. We can employ the same corruption detection idea for domain detection. Since the pre-trained model does not need to be re-trained in our framework, it might be interesting to adopt our framework for domain generalization. For instance, a natural image and cartoon have distinguishable features, such as color distributions, Fourier spectrum, etc. Accurate domain detection might be a simple task if proper features are found.

Currently, our framework accuracy is bounded by the BN statistics update proposed in (Benz et al., 2021; Schneider et al., 2020). As a result, with the presence of perfect corruption/domain detection, the accuracy may not be improved if the BN statistic update does not work for the target corruption/domain. In the future, we will investigate other approaches to eliminate this limitation.

## 5 RELATED WORK

Dodge et al. Dodge & Karam (2017) revealed that deep models' accuracy significantly drops with corrupted images despite having similar performance to humans on clean data. Several studies (Geirhos et al., 2018b; Vasiljevic et al., 2016) verified that training with some corruptions does not improve the accuracy for unseen corruptions. However, Rusak et al. (2020b) later challenged this notion by showing that Gaussian data augmentation can enhance the accuracy of some other corruptions as well. In (Benz et al., 2021; Schneider et al., 2020), authors have shown that corruption accuracy can be significantly increased by only updating the BN statistics of a trained model on a specific corruption. Although it is claimed that it can be easily adopted at inference time by updating the model BN stats using a batch of most recent samples, the performance of the models has not been evaluated in a situation where the corruption type changes.

There are numerous data augmentation methods shown to improve corruption robustness. AutoAugment Cubuk et al. (2019) automatically searches for improved data augmentation policies but has been shown later to improve corruption error Yin et al. (2019). AugMix Hendrycks et al. (2019) combines a set of transforms with a regularization term based on the Jensen-Shannon divergence. It has been shown that applying Gaussian noise to image patches can also improve accuracy Lopes et al. (2019). In Stylized-ImageNet, the idea of using style-transfer were adopted for data augmentation Geirhos et al. (2018a). Using adversarially learned noise distribution has been proposed in Rusak et al. (2020a). In DeepAug Hendrycks et al. (2021), images are passed through image-to-image models while being distorted to create new images leading to large improvements in robustness. The adoption of adversarially training to improve corruption robustness has not been consistent. For instance, Rusak et al. (2020b) has shown that adversarial training does not improve corruption robustness while Shen et al. (2021) and Ford et al. (2019) have reported otherwise, using $l\infty$ adversarial training.

## 6 CONCLUSION

In this paper, we propose a framework where an off-the-shelf naturally trained vision model can be adapted to perform better against corrupted inputs. Our framework consists of three main components: 1) corruption type detector, 2) BN stats lookup table, and 3) an off-the-shelf trained model. Upon detecting the corruption type with the first component, our framework pulls the corresponding BN stats from the lookup table and substitutes the BN stats of the trained model. Then, the original image is fed to the updated trained model.

Even though detecting the corruption type is a very challenging task in the image domain, we can use the Fourier spectrum of an image to detect the type of corruption. We use a shallow three-layer FC neural network that detects the corruption type based on Fourier amplitudes of the input. We show that this model can achieve significant accuracy by training on minimal samples. The same small sample size is shown to be also enough to obtain the BN stats stored in the BN stat lookup table.

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
