# OpenReview forum: "DYNAMIC BATCH NORM STATISTICS UPDATE FOR NATURAL ROBUSTNESS"
_ICLR.cc/2023/Conference — Submitted to ICLR 2023_

### Official Review · Reviewer_s7TA · 2022-10-23

**Confidence:** 4
**Correctness:** 3
**Technical Novelty And Significance:** 2
**Empirical Novelty And Significance:** 2
**Recommendation:** 3

**Clarity, Quality, Novelty And Reproducibility:**

Clarity & Quality
- The main component, "Inference Time Adaptation," is not in the method section (Section 2), but in the experiment section (Section 3.4). For a better understanding of the reader, I think it has to be included in the method section. Also, it is unclear how it works "We let the model BN stats be updated from the last ten batches at the beginning of each batch"
- Explanations for the BN lookup table are not enough, as I mentioned above.
- The evaluation protocol for dynamic corruption is ambiguous, as I mentioned above.

Novelty
- I think the technical novelty is limited. The benefits of Fourier-based and BN-based approaches are not new. I agree with the importance of the tackled problem, but the proposed method is limited in many perspectives, e.g., the use of corruption supervision, without considering unseen corruption type at the inference.

Reproducibility
- I did not find any code implementations.



**Strength And Weaknesses:**

Strengths
- The writing is clear and easy to understand
- This manuscript introduces more realistic scenarios, which handle dynamic corruption during the testing phase, than the prior works
- The experimental results are strong compared to the baseline.

Weaknesses
- Strong assumptions and Gaps between the tackled problem and the proposed method; To overcome the drawback of the prior works, which assume a single corruption type at the inference phase, the authors target the case dynamic corruption types given at the inference phase. However, as the authors discussed in the Limitation section, the proposed method requires several strong assumptions on the problem; the authors assume that (1) they have all the corruption supervision at the training phase to train the detector model, (2) there are only seen corruptions at the inference phase, and (3) the testing samples are sorted in order to have the same corruption types in consecutive batches, not randomly at the inference phase. Furthermore, guessing from the lack of explanation, it seems to assume to know when the corruption types change and know what it is at the inference phase without using the detector. This doubt comes from Figure 6, which shows the fixed performance over different testing setups despite the performance of the detector is not high (nearly 50% on CIFAR-10C).
- Weak technical novelty; all components of the proposed method are from the literature [1,2,3]. The benefits of using the Fourier domain or solely updating BN statistics for robustness are empirically known in the recent literature. The authors also just followed BN updating rules from the literature.
- Weak baselines; overall, this manuscript does not provide enough baselines to evaluate the significance of the proposed method. There are no baselines handling robustness (e.g., [1,2]) in Tables 2 and 3 (which show the effectiveness of the single corruptions given at the inference). Similarly, it is difficult to understand the effectiveness of the proposed Fourier detector; Are the accuracies of 49% on CIFAR-10C and 65% on ImageNet-C high? There are no comparisons between Fourier-based methods like [1].
- Lack of explanation: There are no clues on how to obtain the BN lookup Table. How many samples are used to obtain the corruption-specific BNs? How do we use the trained detector to obtain the lookup Table?

[1] Benz et al., "Revisiting batch normalization for improving corruption robustness," ICCV 2021
[2] Schneider et al., "Improving robustness against common corruptions by covariate shift adaptation," NeurIPS 2020
[3] Yin et al., "A Fourier Perspective on Model Robustness in Computer Vision," NeurIPS 2019


**Summary Of The Paper:**

This manuscript proposes a way to update BatchNorm statistics to improve the robustness of dynamic corruptions at inference time. Specifically, the proposed method detects corruptions at the Fourier domain and adaptively updates BN statistics based on the detected corruption types. The extensive experiments showed the effectiveness of the proposed method on the robustness benchmarks such as CIFAR-10C and ImageNet-C.

**Summary Of The Review:**

Overall, I recommend rejection for the following reasons;
- At the high level, the proposed method is motivated to handle more realistic scenarios on the model robustness, however, it further requires more strict and unrealistic assumptions such as corruption supervision, and all corruption types are known in the training phase.
- The provided explanations and experimental results are not enough to clearly understand the proposed method and its effectiveness (e.g., weak baselines, lack of explanations, as I raised above)
- This work has limited novelty, as I raised above.

---

### Official Review · Reviewer_uu4e · 2022-10-24

**Confidence:** 4
**Correctness:** 4
**Technical Novelty And Significance:** 2
**Empirical Novelty And Significance:** 2
**Recommendation:** 5

**Clarity, Quality, Novelty And Reproducibility:**

The paper is clear. It is of 'medium' quality – see the strengths and weaknesses above. The approach is not particularly novel: as mentioned above, the foundations were set in prior work, including the connection between frequency-domain statistics and corruption types. The paper is sufficiently reproducible, provided that the corruption-specific batch norm stats from prior work are public.

**Strength And Weaknesses:**

The biggest strength of the paper is that it may be practically useful in some settings (in particular those that may be subject to the specific types of corruptions considered (jpeg compression, motion blur, snow, etc.). The authors show that, if these same corruptions are encountered, then this approach can generally improve performance without the need to retrain the original model (which is more difficult to retrain than the small corruption-type classifier). In addition, the authors are transparent about the limitations in Section 4.

The biggest weaknesses of the paper are that 1. insights for the reader are limited and 2. the problem being solved -- mitigating effects only of *known* types of input corruption, while further restricted to be unable to retrain the underlying network -- is too restrictive, and not well motivated. With regard to (1), many interesting bits were already tackled in prior work, e.g. proposing batch norm updates for robustness to corruption (e.g. Benz et al. 2021), showing that frequency magnitude is indicative of corruption type (e.g. Figures 3 and 4 are very similar to this in Yin et al. 2019). The method described for frequency-domain normalization, before the corruption-type classifier, does have merit, but at the same time it is a simple heuristic, and in addition it's never tested through an ablation study. With regard to (2), I admire the authors for highlighting the obvious limitation of 'known corruption' in Section 4, but it is nevertheless a significant limitation. Further, it is not at all clear that the proposed directions for alleviating this limitation would work (outlier detection or unsupervised learning for learning new corruption types), and, even if one could possibly get this to work – wouldn't it be easier in most cases to just retrain, or more realistically fine tune, the underlying models?

**Summary Of The Paper:**

An approach is proposed for improving performance when corruption types are known in advance, and when one cannot or is not willing to retrain the underlying model. The approach involves 1. training a small corruption-type classifier that operates on carefully-normalized, frequency-domain input images, 2. a table of batch norm statistics for each corruption type, and 3. an existing network that heavily leverages batch normalization (this network does not need to be retrained, but rather has its BN stats updated based on the corruption type, as determined by the corruption-type classifier). This method is evaluated only on the same types of corruptions which undergo the same 15 corruption types that were used to train the corruption-type classifier, using CIFAR10-c and ImageNet-C, standard corrupted variants of CIFAR and ImageNet. In this setting, the proposed approach is carried out using a variety of common networks (ResNet-18, ResNet-50, DenseNet, etc.), and shown to generally perform better on corruption inputs (+ approx. 8% for CIFAR, + approx. 4% for ImageNet) and a bit worse on non-corrupted images (- approx. 0.5% for CIFAR, - approx 1% for ImageNet).

**Summary Of The Review:**

The experiments show that the proposed solution 'works', but I can't recommend acceptance primarily because of the weaknesses above, and in particular because of the problem itself. My biggest suggestion to the authors is to try to strongly motivate the problem of mitigating effects of known input corruption types while being unable to retrain the underlying model.

---

### Official Review · Reviewer_1hRd · 2022-10-25

**Confidence:** 4
**Correctness:** 2
**Technical Novelty And Significance:** 2
**Empirical Novelty And Significance:** 2
**Recommendation:** 3

**Clarity, Quality, Novelty And Reproducibility:**

- Clarity: The method is clearly outlined. The results could make use of more summary tables, rather than individual results on the corruptions (which can also be moved to the appendix)
- Quality: The paper needs to be improved, e.g. in terms of the references to back up particular statements (cf. above) and in terms of control experiments.
- Novelty: The technical contribution is incremental and it is likely not able to outperform more recent test-time adaptation methods. While it is (to my knowledge) conceptually new to detect the corruption type in Fourier space, the associated claims are not sufficiently backed up by experiments.
- Reproducibility: There is sufficient detail and reference results to reproduce the experiments.

**Strength And Weaknesses:**

I opted for a relatively short summary of the main weaknesses, as it does not make sense to discuss the paper in more detail before fixing/commenting these, and I am looking forward to discussing with the authors.

**Major Weaknesses**

- The literature is not sufficiently discussed. E.g. the first entry paragraph does not back up any sentence with citations, and the following paragraphs are also quite sparse in citations.
- The paper is lacking a discussion of more powerful methods than Batch norm adaptation: What about more recent test-time adaptation methods like TENT (Wang et al., 2020) or EATA (Niu et al., 2022)?
- "[...] detect the corruption type, a challenging task in the image domain": While I like the authors' approach to use the Fourier spectrum, this sentence seems to be wrong. Any one-layer convnet with sufficiently large kernel can learn to perform a Fourier transform, so arguably also if the "domain detector" is trained in "image space", the network could discover this stategy. The authors should provide experimental evidence that this sentence is true, or remove it from the abstract.
- The proposed method based on the Fourier spectrum is surprisingly bad (cf. Figure 5). Especially the noise types are confused. This shows an inherent limitation of the method, as e.g. distinguishing shot and gaussian noise is trivial in the image domain (shot noise has extreme values at 0/255 and can be easily detected in the histogram).

**Minor Weaknesses:**

- The citations are not correctly typeset (using `\cite` instead of `\citep` at most places)



**Summary Of The Paper:**

The paper builds on previous batch norm adaptation results, in which statistics are computed from a larger batch of test images. Instead, the authors build a "domain predictor" based on frequency features, and then learn conditional statistics for each of the domains. The method is benchmarked on CIFAR-C and ImageNet-C.

**Summary Of The Review:**

While the approach is interesting and might be worth pursuing, the paper lacks rigour in the evaluation and the discussion of the results within the existing literature. Specifically the main claim about the usefulness of the Fourier space to distinguish corruption domains is not sufficiently supported by experiments.

---

### Official Review · Reviewer_Jakz · 2022-10-26

**Confidence:** 3
**Clarity, Quality, Novelty And Reproducibility:** The writing and novelty of this paper…
**Correctness:** 3
**Technical Novelty And Significance:** 1
**Empirical Novelty And Significance:** 1
**Recommendation:** 3

**Strength And Weaknesses:**

Strength:
- The studied problem is of practical importance.

Weaknesses:
- Many state-of-the-art deep networks, such as vision Transformers, do not leverage batch normalization. The proposed method can not be applied to them.
- Most realistic corruption may not be able to be categorized into one of a small number of pre-defined classes.
- The novelty of this paper may be limited. The proposed method is more like an engineering solution, which integrates some existing techniques. The idea of identifying corruption in the frequency domain has been widely explored in the field of low-level vision.
- The writing of this paper requires improvement.

**Summary Of The Paper:**

This paper seeks to improve the robustness of DNNs against common visual corruption, e.g., snow, fog, and blur. The authors propose to identify the type of corruption by leveraging the Fourier spectrum, and update the BN stats with a pre-computed look-up table. The experimental results are provided on top of ResNet.

**Summary Of The Review:**

I think the current paper is not ready to be published on ICLR. See the weaknesses above.

---

### Decision · Program_Chairs · 2023-01-20

**Decision:**

Reject

**Justification For Why Not Higher Score:**

Poor clarity, poor comparisons. No rebuttal.

**Justification For Why Not Lower Score:**

n/a

**Metareview: Summary, Strengths And Weaknesses:**

In this work, the authors propose a new method to improve the robustness of convolutional neural networks but selectively updating the batch normalization statistics to account for visual corruptions. In particular, the authors employ a lookup table to replace the batch normalization statistics to account based on analysis of the Fourier statistics of the images. The authors test variations of various convolutional neural networks on CIFAR10-C and ImageNet-C which contain corruption benchmarks on the respective datasets. Although the reviewers applauded the paper for approaching a topical problem, the reviewers identified several notable weaknesses including (1) the clarity of presentation, (2) insufficient discussion and comparisons against established literature. In addition, the reviewers surfaced concerns about the overall approach being limited to models which employ CNNs over a discrete and fixed set of corruptions. The authors provided no responses to these points so these weaknesses are unaddressed. For these reasons, this paper will not be accepted into this conference.